# The Emerging Role of NANOG as an Early Cancer Risk Biomarker in Patients with Oral Potentially Malignant Disorders

**DOI:** 10.3390/jcm8091376

**Published:** 2019-09-03

**Authors:** Juan C. de Vicente, Tania Rodríguez-Santamarta, Juan P. Rodrigo, Eva Allonca, Aitana Vallina, Anusha Singhania, Paula Donate-Pérez del Molino, Juana M. García-Pedrero

**Affiliations:** 1Department of Oral and Maxillofacial Surgery, Hospital Universitario Central de Asturias (HUCA), C/Carretera de Rubín, s/n, 33011 Oviedo, Asturias, Spain; 2Department of Surgery, University of Oviedo, Asturias 33006, Spain; 3Instituto de Investigación Sanitaria del Principado de Asturias (ISPA), C/Carretera de Rubín, s/n, 33011 Oviedo, Asturias, Spain; 4Instituto Universitario de Oncología del Principado de Asturias (IUOPA), University of Oviedo, Asturias 33006, Spain; 5Department of Otolaryngology, Hospital Universitario Central de Asturias (HUCA), C/Carretera de Rubín, s/n, 33011 Oviedo, Asturias, Spain; 6Ciber de Cáncer (CIBERONC), Instituto de Salud Carlos III, Av. Monforte de Lemos, 3-5, 28029 Madrid, Spain; 7Department of Pathology, Hospital Universitario Central de Asturias (HUCA), IUOPA, C/Carretera de Rubín, s/n, 33011 Oviedo, Asturias, Spain; 8Christian Medical College and Hospital, Ludhiana 141008, Punjab, India

**Keywords:** oral squamous cell carcinoma, oral cancer risk, oral epithelial dysplasia, NANOG, immunohistochemistry

## Abstract

NANOG, a key regulator of pluripotency and self-renewal in embryonic and adult stem cells, is frequently overexpressed in multiple cancers, including oral squamous cell carcinoma (OSCC). It has been frequently associated with poor outcomes in epithelial cancers, and recently implicated in laryngeal tumorigenesis. On this basis, we investigated the role of NANOG protein expression as an early cancer risk biomarker in oral potentially malignant disorders (OPMD), and the impact on prognosis and disease outcomes in OSCC patients. NANOG expression was evaluated by immunohistochemistry in 55 patients with oral epithelial dysplasia, and 125 OSCC patients. Correlations with clinical and follow-up data were assessed. Nuclear NANOG expression was detected in 2 (3.6%) and cytoplasmic NANOG expression in 9 (16.4%) oral dysplasias. NANOG expression increased with the grade of dysplasia. Cytoplasmic NANOG expression and the histopathological grading were significantly correlated with oral cancer risk, although dysplasia grading was the only significant independent predictor of oral cancer development in multivariate analyses. Cytoplasmic NANOG expression was also detected in 39 (31%) OSCC samples. Positive NANOG expression was significantly associated with tobacco and alcohol consumption, and was more frequent in pN0 tumors, early I-II stages. These data unveil the clinical relevance of NANOG in early stages of OSCC tumorigenesis rather than in advanced neoplastic disease. NANOG expression emerges as an early predictor of oral cancer risk in patients with OPMD.

## 1. Introduction

Oral squamous cell carcinoma (OSCC) is a malignancy characterized by genomic instability, cellular heterogeneity, and a dismal prognosis, since more than 50% of patients still die of this disease or complications within 5 years [1]. OSCC may develop from histologically normal oral mucosa or from oral potentially malignant disorders (OPMDs), such as oral leukoplakia (OLK), erythroplakia, or lichen planus, in a process which takes place at the normal epithelium, progressing through hyperplasia to dysplasia and culminating in an invasive carcinoma [2]. OPMDs may show many of the genetic alterations which are present in OSCC [3], even in the absence of histologically defined dysplasia [4], which currently remains the best predictor of progression to invasive carcinoma [5].

In view of an emerging concept of carcinogenesis, the cancer stem cell hypothesis, a subpopulation of cells termed as cancer stem cells (CSCs) or tumor initiating cells (TICs) play a crucial role not only in tumor initiation and maintenance, but also in tumor aggressiveness, microenvironment modulation, evasion of apoptosis, and metastatic spreading [6,7,8,9,10]. The early transcription factors NANOG, OCT4, and SOX2 play pivotal roles in the maintenance of pluripotency and self-renewal capability in both embryonic and adult stem cells. In addition, it has been demonstrated that these factors are key regulators of CSCs properties and self-renewal in head and neck squamous cell carcinomas (HNSCC) [11]. Specifically, NANOG has been shown to be upregulated in different types of cancers including OSCC, and its overexpression has been correlated with poor differentiation status, poor prognosis, and chemoresistance [11,12], suggesting that NANOG may promote aggressive tumor phenotypes [13]. However, Hwang et al. [14] and Vaz et al. [15] respectively found that NANOG expression was not related to prognosis in esophageal and rectal cancers. Therefore, the prognostic value of NANOG expression in solid tumors remains controversial. On the other hand, a recent paper by Rodrigo et al. [16] uncovered a novel role for NANOG in the early stages of laryngeal tumorigenesis, and more importantly, its clinical application as a biomarker for cancer risk assessment in patients with laryngeal precancerous lesions. 

The present study was conducted to comprehensively investigate the clinical relevance of NANOG expression in both early stages of oral carcinogenesis and late stages of disease progression, by analyzing NANOG protein expression using immunohistochemistry in large series of oral dysplastic lesions and OSCC tissue specimens, to establish correlations with the risk of progression to oral cancer, impact on OSCC prognosis and patient outcome.

## 2. Methods and Materials

### 2.1. Patients and Tissue Specimens

Surgical tissue specimens from 55 patients with a diagnosis of oral epithelial dysplasia at the Hospital Universitario Central de Asturias between 2000 and 2005 were retrospectively collected. Patients included in this study had to meet the following criteria: (i) pathological diagnosis of oral epithelial dysplasia; (ii) feature lesions of the oral mucosa (leukoplakia); (iii) no previous history of head and neck cancer, (iv) complete excisional biopsy of the lesion; and (v) a minimum follow-up of five years (or until progression to malignancy occurred). Fifty-five patients who met these criteria were included in this study. Patients were followed-up every two months for the first six months after completing the treatment, every three months until the second year, and every six months thereafter. Representative tissue sections were obtained from archival, paraffin-embedded blocks and the histological diagnosis was confirmed by an experienced pathologist. The premalignant lesions were classified into the categories of low-grade and high-grade dysplasia, following the current WHO classification [17]. Alveolar mucosa obtained from unerupter third molars surgery was used as control. All patients gave their consent to excise this normal tissue.

Additionally, an independent cohort of 125 patients with histologically confirmed OSCC who underwent surgical treatment with curative purposes at the Hospital Universitario Central de Asturias between 1996 and 2007 were retrospectively collected, in accordance with approved institutional review board guidelines. All experimental procedures were conducted in accordance with the Declaration of Helsinki and approved by the Institutional Ethics Committee of the Hospital Universitario Central de Asturias and by the Regional CEIC from Principado de Asturias (date of approval 5 May 2016; approval number: 70/16) for the project PI16/00280. Informed consent was obtained from all patients. Clinicopathologic data were collected from medical records, as summarized in Appendix A. Tissue samples and data from donors included in this study were provided by the Principado de Asturias BioBank (PT17/0015/0023), integrated in the Spanish National Biobanks Network and they were processed following standard operating procedures with the appropriate approval of the Ethical and Scientific Committees. Representative tissue samples were obtained from archival, formalin-fixed paraffin-embedded blocks to construct tissue microarrays.

### 2.2. Tissue Microarray (TMA) Construction

Three morphological representative areas were selected from each individual paraffin block, and 1 mm diameter tissue cores were transferred to the recipient master block to construct the TMAs. The original archived hematoxylin- and eosin-stained slides were reviewed by an experienced pathologist, who identified the areas of interest and confirmed the histological diagnosis. Each TMA block also included three cores of normal epithelium as an internal control. These samples were obtained from non-oncological patients undergoing oral surgery. In order to check the histopathological diagnosis and the adequacy of tissue sampling, a section from each microarray was stained with hematoxylin and eosin and examined by light microscopy.

### 2.3. Immunohistochemistry (IHC)

The TMAs were cut into 3 μm sections and dried on Flex IHC microscope slides (DakoCytomation, Glostrup, Denmark). The sections were deparaffinized with standard xylene and hydrated through graded alcohols into water. Antigen retrieval was performed by heating the sections with Envision Flex Target Retrieval solution, high pH (Dako). Staining was performed at room temperature on an automatic staining workstation (Dako Autostainer Plus, Dako) with NANOG (D73G4) XP^®^ rabbit monoclonal antibody (Cell Signaling technology, Inc.) at 1:200 dilution, using the Dako EnVision Flex + Visualization System (Dako Autostainer, DakoCytomation, Glostrup, Denmark). Negative controls were prepared by omitting the primary antibody. Counterstaining with hematoxylin was the final step. 

The IHC results were independently evaluated by two observers (JPR, and JMG-P), blinded to clinical data. Given that CSC subpopulations could represent a very small percentage of cells, hence NANOG expression in few cells even as low as 1% could be intrinsically meaningful into the CSC concept. Taking this into consideration, any NANOG–positive cell was considered even 1% of positive cells. A semiquantitative scoring system based on staining intensity was applied, as previously established [16], divided into three categories: negative (absence of staining, score 0); weak to moderate (some cytoplasmic staining in dysplastic areas, score 1); and strong protein expression (intense and homogeneous cytoplasmic staining in dysplastic areas, score 2), with an inter-observer concordance higher than 95%. As in some cases nuclear staining was observed, the cases were also scored as positive/negative based on the presence of nuclear staining in dysplastic areas. In OSCC, also a semiquantitative scoring system based on staining intensity was applied as: negative (0), weak (1) and strong protein expression (2), as previously established [18]. Since any NANOG–positive staining could be meaningful, these criteria were used as a cut-off point to establish positive NANOG expression (scores 1 and 2) vs. negative expression (score 0). Human seminoma was used as positive control, showing strong nuclear NANOG staining.

### 2.4. Statistical Analysis

Bivariate analyses by χ^2^ and Fisher’s exact tests were used for comparison between NANOG expression and clinicopathological categorical variables. Disease-specific survival (DSS) was determined for the date of treatment completion to death for the tumor. For time-to-event analysis, survival curves were estimated using the Kaplan–Meier method. The log-rank test was used to compare the survival curves. Hazard ratios (HR), with their 95% confidence intervals (CI) for clinicopathological variables, were calculated using the univariate Cox proportional hazards model. All tests were two-sided and *p* values less than 0.05 were considered statistically significant. All statistical analyses were performed using SPSS version 18 (IBM Co., Armonk, NY, USA).

## 3. Results

### 3.1. Patient Characteristics 

A total of 55 patients diagnosed with oral epithelial dysplasia were selected for study. Twenty-six patients (53%) were men and the remaining 29 women (47%), with a mean age of 62.61 years (SD 12.56, range 39 to 83 years). Regarding tobacco and alcohol consumption, information was only available for 31 patients, and ten of these patients (32%) were smokers and 4 (13%) habitual alcohol drinkers. Forty-two of 55 patients (76%) were classified as low-grade dysplasia, and 13 (24%) as high-grade dysplasia, according to the current WHO classification [17]. 

The clinical and pathological characteristics of the 125 OSCC patients selected for study are shown in Appendix A. This cohort was composed of 82 (66%) men and 43 (34%) women, with a median age of 57 years ranging from 28 to 91 years. Eighty-four patients (67%) were smokers and 69 (55%) were habitual alcohol drinkers. Most of the tumors were well differentiated (64%), more than 50% of cases were in advanced clinical stages (III or IV), and the most common site of tumor origin was the tongue (41%) followed by the floor of the mouth (30%). Neck node metastases were present in 49 (39%) cases, and local recurrences were found in 54 (43%) cases. No patient had distant metastasis at the time of diagnosis. Adjuvant radiotherapy was administered to 75 patients (60%), and adjuvant chemotherapy was administered to 14 patients (11.2%). 

### 3.2. Immunohistochemical Analysis of NANOG Expression in Oral Epithelial Dysplasias

NANOG protein expression was evaluated by immunohistochemistry in a set of 55 oral epithelial dysplasias. Nuclear NANOG expression was detected in 2 (3.6%) cases. Positive cytoplasmic NANOG expression was detected in 9 (16.4%) oral dysplasias: five (9.1%) lesions showed strong staining (score 2) and four (7.2%) lesions weak to moderate staining (score 1). Figure 1 shows representative examples of nuclear/cytoplasmic NANOG expression in oral dysplasias, compared to the negative expression in normal adjacent epithelia (Figure 1A–C). Strong nuclear NANOG staining was detected in human seminoma, used as a positive control (Figure 1D). 

Cytoplasmic NANOG expression was significantly correlated with the histopathological classification. Thus, 4 (10%) of 42 lesions with low-grade dysplasia, and 5 (38%) of 13 lesions with high-grade dysplasia exhibited cytoplasmic NANOG protein expression (Fisher’s exact test *p* = 0.02) (Table 1). Nuclear NANOG expression showed a trend to associate with a higher grade of dysplasia (Fisher’s exact test *p* = 0.05) (Table 1). 

### 3.3. Association of NANOG Protein Expression with Oral Cancer Risk

During the follow-up period, 12 (22%) of 55 patients developed an invasive OSCC at the same site of the previous premalignant lesion. The mean and median times to cancer diagnosis in the cases that progressed were 184 months (range 145 to 222 months) and 192 months (range 24 to 359 months), respectively. The histopathological grade of dysplasia was significantly correlated with the risk of progression from oral epithelial dysplasia to invasive carcinoma in the present cohort (*p* < 0.001; Table 2). In addition, patients harboring NANOG–positive dysplasias either considering cytoplasmic or nuclear NANOG expression were consistently and significantly associated with an increased risk of progression to oral cancer (*p* = 0.02 and *p* = 0.04, respectively) (Table 2). Univariate Kaplan–Meier and Cox analysis also showed that cytoplasmic NANOG, nuclear NANOG, and the histological grade of dysplasia were significantly associated with oral cancer risk (*p* = 0.002, *p* = 0.001 and *p* < 0.001, respectively) (Table 3 and Figure 2). When these three factors were simultaneously analyzed using a multivariate Cox analysis, dysplasia grading was the only significant independent predictor of oral cancer development (HR = 17.88, 95% CI 3.59 to 89.04; *p* < 0.001) (Table 4). In addition, patients carrying strong cytoplasmic NANOG expression (score 2) experienced a higher progression to OSCC than those with negative to moderate (scores 0 and 1) expression, and these differences almost reached statistical significance (HR = 4.35, 95% CI 0.88 to 22.42; *p* = 0.07). 

### 3.4. Clinical Significance of NANOG Protein Expression in OSCC Progression and Disease Outcome

NANOG protein expression was evaluated by immunohistochemistry in a cohort of 125 OSCC patients (Appendix A). Positive NANOG expression (scores 1 and 2) was detected in 39 (31%) of 122 carcinomas (3 cases were not evaluable); cytoplasmic staining was predominantly observed in tumor cells, and negligible staining in stromal cells (Figure 1E,F). Regarding the clinicopathological variables, positive NANOG expression was significantly correlated with smoking habit (*p* = 0.009), and alcohol consumption (*p* = 0.01) (Table 5). Even though no other significant correlations were observed, positive NANOG expression was more frequent in pN0 tumors, early I-II stages, and absence of tumor recurrences (Table 5). 

Over a median follow-up of 61 months (range, 1 to 230 months), 53 deaths occurred. The mean and median follow-up times were 71.82 (SD: 57.55), and 61.0, respectively. The 5- and 10-year disease-specific survival rates were 60% and 44%, respectively. The mean and median survival times were 132.74 months (95% CI: 113.25 to 152.22 months), and 141 months (95% CI: 102.40 to 179.59 months), respectively. In the survival analyses, tumor size and local extension (T), neck node status (N), and clinical stage were significantly correlated to survival (*p* = 0.001, *p* = 0.01, and *p* = 0.002, respectively). Positive NANOG expression was associated with a higher 5-year disease-specific survival, although this relationship did not reach statistical significance (*p* = 0.389) (Figure 2). 

### 3.5. In Silico Analysis of NANOG and OCT4 mRNA Expression Using TCGA Data

The role of NANOG was further investigated by a transcriptomic analysis of RNAseq data available from The Cancer Genome Atlas (TCGA) HNSCC cohorts [19], using the platforms cBioPortal (http://cbioportal.org/) [20] and UALCAN (http://ualcan.path.uab.edu/) [21]. Thus, analysis of an extended TCGA cohort of 530 HNSCC patients showed that NANOG mRNA levels significantly increased in primary tumors compared to normal tissue samples (*p* < 0.001, Figure 3A), in agreement with our results at protein level. Similarly, OCT4 mRNA levels were found to significantly increase in primary tumors compared to normal tissues (*p* <0.001, Figure 3A). OCT4 is an important CSC regulator functionally related to NANOG and also a transcription factor known to regulate NANOG expression. We next assessed the alteration frequency of NANOG mRNA and other CSC-related genes (i.e., OCT4, SOX2 and Podoplanin, PDPN) specifically in the subset of 172 OSCC patients from the TCGA cohort. As shown by heatmap analysis (Figure 3B), NANOG and OCT4 mRNA levels were found to be upregulated in 5 (2.9%) and 3 (1.7%) cases, respectively. Moreover, concomitant NANOG and OCT4 expression was only detected in 1 case (0.6%), and as such extremely rare in OSCC patients. These data indicate that NANOG expression is not frequently altered at transcriptional level. In addition, when evaluating the impact of NANOG mRNA expression on OSCC patient survival, patients carrying NANOG up-regulation exhibited higher survival, although statistical significance was not attained (*p* = 0.483, Figure 3C). Up-regulation of SOX2 and PDPN mRNAs was detected in 22 (13%) and 6 (3%) cases, respectively (Figure 3B). Notably, there was almost no overlap in alterations between all four CSC-related genes in this subset of 172 OSCC patients. The correlation between NANOG and PDPN protein expression was also assessed in 26 OPMDs with data available (Appendix A). We found a significant inverse association between NANOG and PDPN proteins in OPMDs (Chi square test, *p* = 0.017). 

## 4. Discussion

This study is the first to demonstrate the clinical significance of NANOG expression in early stages of oral tumorigenesis. Cytoplasmic and nuclear NANOG expression was detected early in oral epithelial dysplasias while being absent in normal adjacent epithelia, and positive NANOG expression in oral dysplasias was significantly correlated with a higher risk of progression to invasive carcinoma. In multivariate Cox analysis, histopathological grading was the only significant independent predictor of oral cancer development in our series; however, patients harboring lesions with strong NANOG expression clearly showed a higher risk of progression (HR >4), almost reaching statistical significance (*p* = 0.07). Nevertheless, since NANOG expression was only detected in 5 out of 12 OPMDs that subsequently progressed to carcinoma, this suggests that NANOG seems to partially contribute as a driver gene to promote OSCC tumorigenesis. Alternatively, spatial–temporal reasons could also explain the lack of NANOG expression in OPMD that lately progressed to OSCC, as plausibly, OPMDs could have been biopsied before aberrant NANOG expression occurred, or cancer could develop from lesions not clinically visible at the time of biopsy and consequently unexamined.

The histological grade of epithelial dysplasia in OPMDs is still currently used as the best predictor of progression to cancer [5]. However, the accuracy of the grading system is largely subjective and affected by a great inter-examiner and intra-examiner variability [22]. The identification of better and more accurate biomarkers capable of robustly predicting the malignant transformation of OPMDs therefore emerges as a valuable strategy to counteract these limitations [23]. In this sense, we and others have contributed to identify various cancer risk biomarkers that exhibited higher predictability beyond dysplasia grading, such as Cortactin (CTTN), the focal adhesion kinase (FAK) and Podoplanin (PDPN) that were strong independent predictors of oral cancer risk but not histopathological diagnosis [24,25]. Noteworthy, the WHO three-tier grading was used in all these studies, while we have used the new binary grading system (high-grade vs low-grade) proposed in the 2017 fourth edition of the WHO Classification. This could also play a role in the differences of predictive values observed with the present study.

Recent advances in next-generation sequencing (NGS) and omics technologies have enormously contributed to uncovering the high complexity and heterogeneity of oncogenomes [26,27]. Beyond the great diversity of genetic and epigenetic alterations found within a tumor, the interaction with the surrounding microenvironment also dynamically modulates the tumor heterogeneity [28]. In addition, epithelia to mesenchymal transition and CSC plasticity has also been demonstrated to fuel tumor heterogeneity in response to environmental cues to drive tumor spreading and therapeutic resistance [29,30]. The fact that OSCC show a heterogeneous architecture compared with healthy oral mucosa has also led to the hypothesis that only a small, clonogenic subpopulation of cells considered CSC or tumor-initiating cells (TICs) is responsible for generating tumors [31]. In this regard, PDPN has been identified as a marker for tumor-initiating cells (TIC) in squamous cell carcinomas [32], and PDPN-positive cells beyond the basal layer of the oral epithelium have been interpreted as an upward clonal expansion of stem cells during carcinogenesis [33]. Consistent with this role, PDPN-positive OPMDs harboring such clonal expansion exhibited a significantly higher risk of malignant transformation, as we and others demonstrated [25,33,34]. 

NANOG is a transcription factor that plays a critical role during embryonic development and is a key regulator of pluripotency in both embryonic stem cells [35,36], adult stratified epithelia, including oral mucosa [18]. Together with other transcription factors such as OCT4 and SOX2 that mediate embryonic stem cell self-renewal, NANOG is down-regulated via hypermethylation during differentiation in embryonic cells [37]. Interestingly, it has been shown that NANOG is required for attaining a pluripotent ground state in the final phase of reprogramming when other key factors are already present and may be fulfilled by activation of NANOG [38]. NANOG is one of the primary downstream targets of OCT4, but the expression of NANOG can also be sustained in the absence of OCT4 [35]. The discovery of downstream regulatory pathways mediated by NANOG indicates that it regulates several biological processes implicated in cancer development, such as self-renewal, tumor cell proliferation, motility, epithelial-mesenchymal transition, escape from the immune system, and drug resistance, which are all defined features for CSCs [39,40]. The majority of cancer cells in a tumor are non-tumorigenic, and therapeutic strategies targeting these cells may cause tumor regression. However, if therapy fails to target the subpopulations of tumorigenic CSCs within a tumor, these cells could regenerate the tumor after treatment, thereby contributing to the appearance of tumor recurrence or metastatic dissemination. Hence, complete eradication of cancers requires the effective targeting and elimination of the CSC subpopulations [39]. CSCs may arise from normal adult epithelial stem cells, and the displacement of normal stem cells by CSCs could be associated with the development of oral field cancerization [2]. OCT4 and NANOG are two of the four factors that give rise to the reprogramming of human somatic cells into germ-line-competent induced pluripotent stem (iPS) cells [41,42]. Functionally, NANOG blocks differentiation [43], and in a comparative analysis focused on the self-renewal of embryonic stem cells, among 17,342 genes, NANOG was ranked within the top 1% [44]. 

NANOG has been found overexpressed in various human cancers, including head and neck squamous cell carcinomas [13]. It has also been recently implicated in early stages of laryngeal carcinogenesis [16]. In agreement with our results, NANOG expression was also revealed as a strong significant predictor of laryngeal cancer risk in patients with precancerous lesions, beyond histological grading [16]. Altogether these data unveil the great applicability potential of NANOG expression as an early cancer risk biomarker, commonly in epithelial premalignancies at different head and neck subsites. Accordingly, since immunohistochemical NANOG evaluation is a relatively simple and objective method that could be easily implemented in the clinical practice, quite reasonably, it emerges as a valuable complementary marker jointly with histological grading for cancer risk assessment. Nevertheless, routine implementation of this molecular test will require further confirmation of these results in future large-sample prospective studies [16].

On the other hand, various studies support the notion that NANOG is highly expressed in late-stage, poorly differentiated, and metastatic carcinomas [13,45,46,47]. However, even though these data suggest that high levels of NANOG are associated with aggressive tumor phenotypes, there are conflicting results on the possible prognostic relevance. In this regard, to further and significantly extend our data in OPMD, this study also included the analysis of NANOG protein expression in a large cohort of 125 OSCC patients homogeneously diagnosed and treated using surgery at the same institution. Positive NANOG expression was detected in 39 (31%) OSCC tissue samples and was significantly associated with tobacco and alcohol consumption. To our knowledge, this is the first study to report a correlation between NANOG expression with smoking and alcohol-drinking habits in cancer, which uncovers a potential relationship between stemness, by means of NANOG as a CSC marker, and classical chemical carcinogens in OSCC. Lee et al. [11] observed a positive correlation between high NANOG expression and mutant p53, and it is well known that p53 is typically mutated in smoking-related cancers. These observations suggest that alcohol and tobacco consumption could trigger oral cavity carcinogenesis, regulating the expression/function of CSC regulatory factors such as NANOG. In this sense, it has been demonstrated that nicotine induced the expression of various CSC markers NANOG, OCT4, CD44 and BMI1, and enhanced CSC properties and tumorigenic potential in HNSCC models in vitro and in vivo [48]. Nevertheless, future studies are needed to deeply investigate this possibility and the underlying mechanisms.

No other significant correlations were observed in our series between NANOG expression with known prognostic factors, such as clinical stage, tumor size or neck lymph node metastasis. Similarly to our results, Rasti et al. [39] reported that NANOG expression did not correlate with any clinicopathological parameters in renal cell carcinomas, although conversely, cytoplasmic NANOG expression was significantly associated to lower survival rates. The impact of cytoplasmic NANOG expression on the disease-specific survival was also assessed in the present study. Kaplan–Meier analysis showed a tendency, but not significant, between positive NANOG expression and better survival in our cohort of OSCC patients. We obtained analogous findings by analyzing NANOG mRNA levels in an independent cohort of 172 OSCC from the TCGA cohort. Thus, higher survival was also observed in patients harboring NANOG mRNA up-regulation. Consistent with this, we also found that positive NANOG expression was more frequent in pN0 tumors, early I-II stages, and the absence of tumor recurrences. In line with our findings, high expression of OCT4 and SOX2 has been associated with earlier stage, small tumor size, and the absence of lymph node metastasis, and high SOX2 expression was significantly associated with better disease-specific survival in OSCC [1]. In addition, we also found that NANOG expression was more frequent in moderately and poorly differentiated OSCC than in well-differentiated tumors, but statistical significance was not reached. These findings may reflect the pluripotency of CSCs and invasive cancer cells [49,50], showing that NANOG expression is not only restricted to CSCs, but also undifferentiated and highly proliferative cells. Thus, NANOG–negative tumors may contain a limited number of undifferentiated OSCC cells, including CSCs [46]. This indicates that NANOG could also be important for the maintenance of an undifferentiated state of malignant cells and development of resistance to therapy [46].

Possible explanations for the contradictory data on the prognostic significance of NANOG protein expression may include methodological differences related to the immunohistochemical NANOG evaluation (i.e., different antibodies and staining scoring system), heterogeneity of patient populations or morphological and genetic heterogeneity in different solid tumors. According to the evidences herein presented, it is also plausible that these discrepancies could underscore the prominent role of NANOG expression in early stages of oral tumorigenesis as a tumor-initiating factor, rather than as a prognostic factor in advanced stages of the neoplastic disease.

In silico analysis of the transcriptome data from the TCGA [19] further contributed to demonstrate the up-regulation of NANOG mRNA expression and other CSC-related genes in OSCC patients. Furthermore, these data provided valuable mechanistic information, as it follows: (*i*) NANOG mRNA up-regulation is detected in OSCC although at much lower frequency than NANOG protein expression, suggesting the involvement of post-transcriptional mechanisms rather than OCT4-dependent transcriptional regulation; (*ii*) patients carrying NANOG mRNA up-regulation exhibited higher survival, as likewise observed for the patients with NANOG–positive expression in our OSCC cohort; (*iii*) mRNA levels of other CSC-related genes OCT4, SOX2 and PDPN were found to be consistently up-regulated in OSCC patients, although these alterations did not overlap; (*iv*) NANOG and PDPN protein expression in OPMDs also showed no overlap despite the expression of each protein significantly predicted an increased risk of malignant progression, thereby suggesting an independent role of these proteins as CSC or TIC during oral carcinogenesis.

## 5. Conclusions

Taken together, this study provides original evidence demonstrating the early occurrence and clinically relevant role of NANOG expression in oral tumorigenesis, rather than in late stages of OSCC progression or patient prognosis. Remarkably, our findings uncover the potential application of NANOG expression as an early predictor of oral cancer risk in patients with oral potentially malignant disorders.

## Figures and Tables

**Figure 1 jcm-08-01376-f001:**
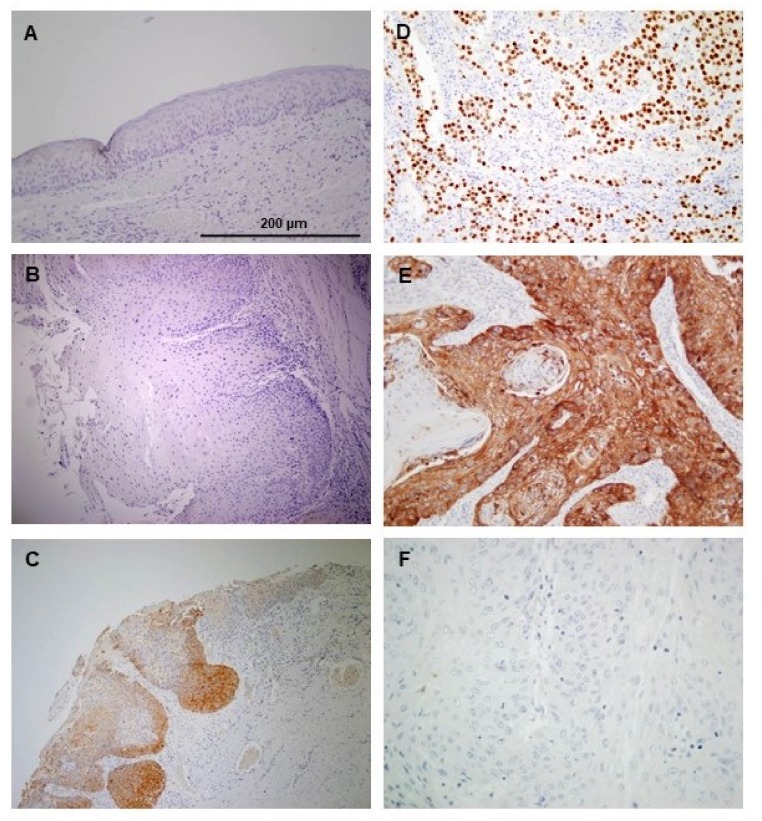
Immunohistochemical analysis of NANOG expression in oral epithelial dysplasias and oral squamous cell carcinoma (OSCC). The normal adjacent epithelium exhibited negative staining (**A**). Representative examples of oral dysplasias showing negative (**B**), and positive NANOG staining (**C**), human seminoma as a positive control (**D**). Examples of oral squamous cell carcinomas with positive (**E**), and negative NANOG staining (**F**). Magnification 200×. Scale bar 200 µm.

**Figure 2 jcm-08-01376-f002:**
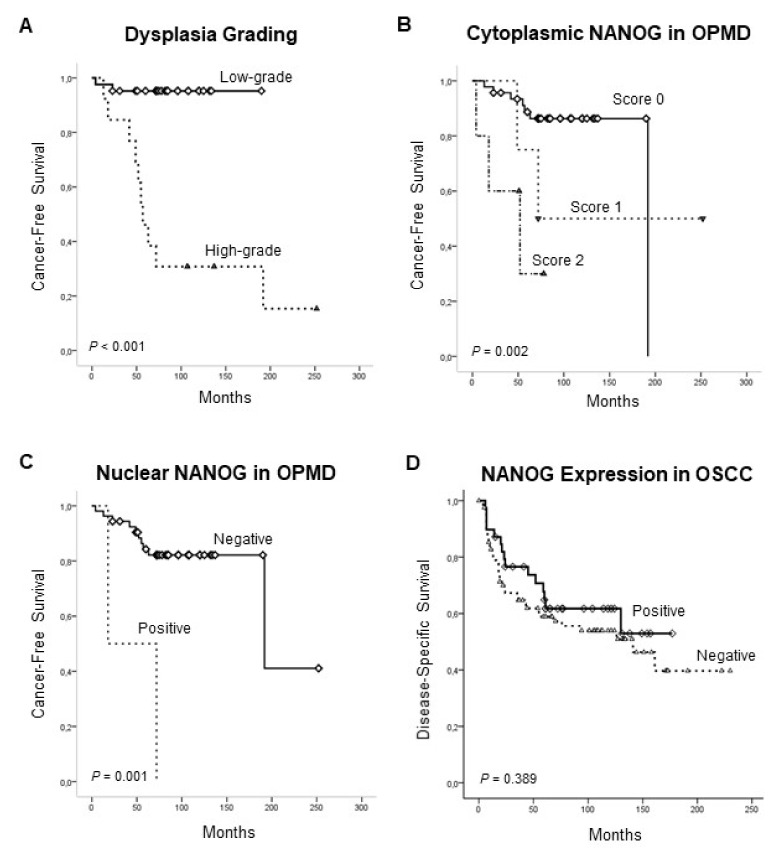
Kaplan–Meier cancer-free survival curves in the cohort of 55 patients with oral epithelial dysplasia categorized by histological grading (low-grade vs. high-grade) (**A**), cytoplasmic NANOG (Staining scores 0, 1 and 2) (**B**) and nuclear NANOG expression (positive vs. negative) (**C**). Disease-specific survival curves in the cohort of 125 OSCC patients dichotomized according to NANOG expression (positive vs. negative) (**D**).

**Figure 3 jcm-08-01376-f003:**
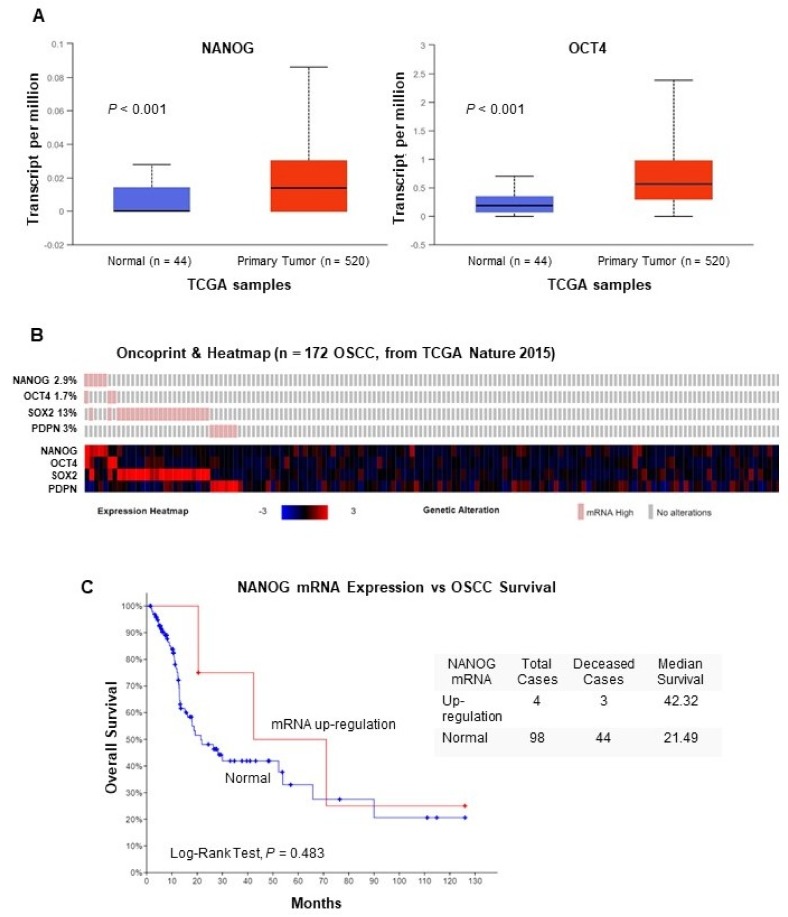
Analysis of NANOG and OCT4 mRNA expression using RNAseq data from the The Cancer Genome Atlas (TCGA) head and neck squamous cell carcinomas (HNSCC) cohorts. (**A**) Box plots comparing the mRNA expression levels of NANOG and OCT4 in primary tumors (in red) VS. normal tissue (in blue) in the TCGA cohort of 530 HNSCC patients using the UALCAN online resources (http://ualcan.path.uab.edu/). (**B**) Oncoprint and heatmap representations showing the percentage of cases with mRNA upregulation of each CSC-related gene assessed in the subset of 172 OSCC patients from the TCGA HNSCC cohort [19]. (**C**) Kaplan–Meier survival curves categorized by NANOG mRNA expression (RNA seq V2 RSEM, *z*-score threshold ±2) dichotomized as normal vs. upregulation, *P* value estimated using the Log-rank test.

**Table 1 jcm-08-01376-t001:** Associations between NANOG protein expression and clinicopathological features in patients with oral dysplasia.

Characteristic	Cytoplasmic NANOG Protein Staining Scores			*p* *	Nuclear NANOG Expression Negative Positive		*p* ^†^
	0	1	2				
Age (years), Mean (SD)	63 (12)	57 (18)	65 (14)	0.69	62 (13)	69 (13)	0.46
Gender, number (%)							
Female	26 (90)	1 (3)	2 (7)	0.42	28 (97)	1 (3)	
Male	20 (77)	3 (11)	3 (11)		25 (96)	1 (4)	
Smoking, number (%)							
Yes	7 (70)	1 (10)	2 (20)	1.00	10 (100)	0 (0)	1.00
No	16 (76)	2 (10)	3 (14)		19 (90)	2 (10)	
Ethanol intake, number (%)							
Yes	1 (25)	2 (50)	1 (25)	0.02	4 (100)	0 (0)	1.00
No	22 (81)	1 (4)	4 (15)		25 (93)	2 (7)	
Dysplasia grading							
Low-grade	38 (90)	1 (3)	3 (7)	0.02	42 (100)	0 (0)	0.05
High-grade	8 (62)	3 (23)	2 (15)		11 (85)	2 (15)	

* Chi square and ^†^ Fisher’s exact tests. Data on tobacco and alcohol intake was only available for 31 patients.

**Table 2 jcm-08-01376-t002:** Evolution of the premalignant lesions in relation to histopathological diagnosis, and nuclear and cytoplasmic NANOG expression.

Characteristic	Number of Cases (%)	Progression to Carcinoma (%)	*p*
Dysplasia Grade			<0.001 ^†^
Low-grade	42 (76)	2 (5)	
High-grade	13 (24)	10 (77)	
Cytoplasmic NANOG			0.02 *
Score 0	46 (84)	7 (15)	
Score 1	4 (7)	2 (50)	
Score 2	5 (9)	3 (60)	
Nuclear NANOG expression			0.04 ^†^
Negative	53 (96)	10 (19)	
Positive	2 (4)	2 (100)	

* Chi square and ^†^ Fisher’s exact tests.

**Table 3 jcm-08-01376-t003:** Univariate Kaplan–Meier and Cox cancer-free survival analysis in 55 patients with oral dysplasias categorized by dysplasia grading, and cytoplasmic and nuclear NANOG expression.

Characteristic	No Cases	Censored Patients (%)	Mean Cancer-Free Survival Time (95% CI)	*p*	Hazard Ratio	95% Confidence Interval
Dysplasia Grade						
Low-grade	42	40 (95)	181.59 (170.21–192.98)	<0.001	Reference	
High-grade	13	3 (23)	100.69 (54.14–147.24)		19.08	4.09–89.01
Cytoplasmic NANOG						
Score 0	46	39 (85)	171.57 (155.07–188.07)	0.002	Reference	
Score 1	4	2 (50)	156.25 (62.07–250.43)		2.30	0.41–12.86
Score 2	5	2 (40)	43.40 (17.52–69.27)		8.13	2.02–32.64
Nuclear NANOG						
Negative	53	43 (81)	189.58 (150.36–228.81)	0.001	Reference	
Positive	2	0 (0)	45.00 (0.00–97.92)		8.13	1.78–38.79

*p* Values were estimated using the log-rank test. 95% CI: 95% Confidence Interval.

**Table 4 jcm-08-01376-t004:** Multivariate Cox proportional hazards model to estimate oral cancer risk.

Variable	*p*	Hazard Ratio	95% Confidence Interval
Histology (High-grade vs. low-grade dysplasia)	<0.001	17.88	3.59–89.04
Cytoplasmic NANOG	0.082		
Score 0	Reference	Reference	
Score 1	0.54	0.55	0.08–3.63
Score 2	0.07	4.45	0.88–22.42
Nuclear NANOG (positive vs. negative)	0.48	2.014	0.28–14.25

**Table 5 jcm-08-01376-t005:** The relationship between clinicopathological variables and NANOG expression in the cohort of 125 OSCC patients.

Variable	No Cases	Positive NANOG Expression (%)	*p*
Gender			
Men	79	31 (39)	0.02
Women	43	8 (18)	
Tobacco use			
Smoker	82	33 (40)	0.005
Non-smoker	40	6 (15)	
Alcohol use			
Drinker	67	29 (43)	0.003
Non-drinker	55	10 (18)	
pT			
pT1 + 2	79	26 (33)	0.76
pT3 + 4	43	13 (30)	
pN			
pN0	75	26 (35)	0.41
pN+	47	13 (28)	
Clinical stage			
I + II	51	20 (39)	0.14
III + IV	71	19 (27)	
G status			
G1	77	21 (27)	0.14
G2 + G3	45	18 (40)	
Tumor location			
Tongue	50	14 (28)	0.43
Other sites	72	25 (35)	
Tumor location			
Floor of the mouth	36	13 (36)	0.52
Other sites	86	26 (30)	
Tumor recurrence			
No	68	26 (38)	0.09
Yes	54	13 (24)	
Second primary carcinoma			
No	104	32 (31)	0.49
Yes	18	7 (39)	
Clinical status at the end of the follow-up			
Alive without recurrence	50	19 (38)	0.48 *
Dead of index cancer	53	15 (28)	
Censored	19	5 (26)	

Fisher’s exact and * Chi-square tests.

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
