# Peer review of "The Emerging Role of NANOG as an Early Cancer Risk Biomarker in Patients with Oral Potentially Malignant Disorders"

_jcm, 2019, doi:10.3390/jcm8091376_

Round 1

Reviewer 1 Report

The paper entitled " Emerging role of NANOG as an early cancer risk  biomarker in patients with oral potentially malignant  disorders" represents an interesting original work to unveil the clinical relevance of NANOG in early stages of OSCC tumorigenesis rather than in advanced neoplastic disease. And NANOG expression as an early biomarker of  oral cancer .

The paper is well written and clear.

“Introduction” seems to be adequate in content and length. It is clear and well written.

“Materials and methods” section seems appropriate and exhaustively defined. The details provided are adequate to replicate the work

“Results” are well written.

Discussion and conclusions are well balanced and adequately supported by the data. They are proportionally developed and clearly written. I suggest to refer also to the following works, to complete exhaustively the discussion section on previous studies:

- della Vella F, Lauritano D, Lajolo C, Lucchese A, Di Stasio D, Contaldo M, Serpico R, Petruzzi M. The Pseudolesions of the Oral Mucosa: Differential Diagnosis and Related Systemic Conditions. Sci. 2019, 9, 2412; doi:10.3390/app9122412

Lauritano D, Lucchese A, Contaldo M, Serpico R, Lo Muzio L, Biolcati F, Carinci F. Oral squamous cell carcinoma: diagnostic markers and prognostic indicators. J Biol Regul Homeost Agents. 2016;30(2 Suppl 1):169-76.

Pannone G, Santoro A, Feola A, Bufo P, Papagerakis P, Lo Muzio L, Staibano S, Ionna F, Longo F, Franco R, Aquino G, Contaldo M, De Maria S, Serpico R, De Rosa A, Rubini C, Papagerakis S, Giovane A, Tombolini V, Giordano A, Caraglia M, Di Domenico M. The role of E-cadherin down-regulation in oral cancer: CDH1 gene expression and epigenetic blockage. Curr Cancer Drug Targets. 2014;14(2):115-27.

Contaldo M, di Napoli A, Pannone G, Losito S, Aquino G, Franco R, Ionna F, Feola A, De Rosa A, Santoro A, Sbordone C, Longo F, Pasquali D, Loreto C, Ricciardiello F, Esposito G, D’Angelo L, Itro A, Bufo P, Tombolini V, Serpico R, Di Domenico M. Prognostic implications of node metastatic features in OSCC: a retrospective study on 121 neck dissections. Oncol Rep. 2013;30(6):2697-704. doi: 10.3892/or.2013.2779.

I recommend that the paper be revised, as suggested, consisting in enriching the "discussion" about previous similar researches, supported by the suggested references and/or other ones.    

Author Response

Reviewer #1

Comments and Suggestions for Authors

The paper entitled "Emerging role of NANOG as an early cancer risk biomarker in patients with oral potentially malignant disorders" represents an interesting original work to unveil the clinical relevance of NANOG in early stages of OSCC tumorigenesis rather than in advanced neoplastic disease. And NANOG expression as an early biomarker of oral cancer.

The paper is well written and clear.

“Introduction” seems to be adequate in content and length. It is clear and well written.

 “Materials and methods” section seems appropriate and exhaustively defined. The details provided are adequate to replicate the work

“Results” are well written.

Discussion and conclusions are well balanced and adequately supported by the data. They are proportionally developed and clearly written.

Response: We thank the reviewer for her/his positive comments and meticulous revision of the manuscript.

Point 1: I suggest to refer also to the following works, to complete exhaustively the discussion section on previous studies:

- della Vella F, Lauritano D, Lajolo C, Lucchese A, Di Stasio D, Contaldo M, Serpico R, Petruzzi M. The Pseudolesions of the Oral Mucosa: Differential Diagnosis and Related Systemic Conditions. Sci. 2019, 9, 2412; doi:10.3390/app9122412

- Lauritano D, Lucchese A, Contaldo M, Serpico R, Lo Muzio L, Biolcati F, Carinci F. Oral squamous cell carcinoma: diagnostic markers and prognostic indicators. J Biol Regul Homeost Agents. 2016;30(2 Suppl 1):169-76.

- Pannone G, Santoro A, Feola A, Bufo P, Papagerakis P, Lo Muzio L, Staibano S, Ionna F, Longo F, Franco R, Aquino G, Contaldo M, De Maria S, Serpico R, De Rosa A, Rubini C, Papagerakis S, Giovane A, Tombolini V, Giordano A, Caraglia M, Di Domenico M. The role of E-cadherin down-regulation in oral cancer: CDH1 gene expression and epigenetic blockage. Curr Cancer Drug Targets. 2014;14(2):115-27.

- Contaldo M, di Napoli A, Pannone G, Losito S, Aquino G, Franco R, Ionna F, Feola A, De Rosa A, Santoro A, Sbordone C, Longo F, Pasquali D, Loreto C, Ricciardiello F, Esposito G, D’Angelo L, Itro A, Bufo P, Tombolini V, Serpico R, Di Domenico M. Prognostic implications of node metastatic features in OSCC: a retrospective study on 121 neck dissections. Oncol Rep. 2013;30(6):2697-704. doi: 10.3892/or.2013.2779.

Response 1: Following the reviewer’s recommendation, additional information and references have been added to the Discussion. We have carefully checked the suggested papers, and one of them has been included (new Ref. 23) together with other publications that we found more closely related to our study, and consequently more adequate to be cited in this revised version of the manuscript.

Point 2: I recommend that the paper be revised, as suggested, consisting in enriching the "discussion" about previous similar researches, supported by the suggested references and/or other ones.

Response 2: The Discussion has been further extended and enriched with additional data on previous related studies investigating the role of CSC markers in OSCC.

Reviewer 2 Report

In the present manuscript, the authors determined the roles of NANOG in tumorigenesis and progression of oral squamous cell carcinoma (OSCC). Using immunohistochemistry in 55 patients with oral potentially malignant disorders (OPMDs) and 125 OSCC patients, the authors showed that the expression of NANOG in OPMD tissues is correlated with progression of OPMD to carcinoma. Positive NANOG expression is also associated with tobacco and alcohol consumption. The authors think the data should reveal the clinical relevance of NANOG in OSCC tumorigenesis rather than in the disease progression. Overall, this is an interesting paper to provide the evidence of the potential relevance of NANOG in OSCC formation, especially the examination of NANOG expression in 12 of 55 patients developed an invasive OSCC. However, there are several major issues that the authors need to address:

The scoring system of IHC used in this study is not convincing. The intensity of staining is not included and considered. The NANOG expression in tissues has to be evaluated with better scoring system, such as H-score system. In Table 1, the total patient numbers of several characteristics are not 55. If the information is not available, it has to be noted below. The authors conclude that the expression of NANOG in OPMD tissues is correlated with progression of OPMD to carcinoma (Table 3). However, for 12 OSCC tissues derived from the OPMDs, only 5 OPMD tissues express the NANOG proteins (Table 2). And the NANOG protein can be detected in only 31% of OSCC tissues (Table 5), suggesting that the expression of NANOG may play a minor role in progression of OPMD to carcinoma. This likely explains that only the dysplasia grading was significant independent predictor of oral cancer development in the multivariate Cox analysis (Table 4), while one would expect the NANOG level is also significant. The rationale for selecting NANOG in OSCC research may be not sufficient. If the NANOG is selected based on the cancer stem cell (CSC) hypothesis, the survey of other CSC-related molecules in OPMD and/or OSCC tissues should be included in the study.

Author Response

Reviewer #2

Comments and Suggestions for Authors

In the present manuscript, the authors determined the roles of NANOG in tumorigenesis and progression of oral squamous cell carcinoma (OSCC). Using immunohistochemistry in 55 patients with oral potentially malignant disorders (OPMDs) and 125 OSCC patients, the authors showed that the expression of NANOG in OPMD tissues is correlated with progression of OPMD to carcinoma. Positive NANOG expression is also associated with tobacco and alcohol consumption. The authors think the data should reveal the clinical relevance of NANOG in OSCC tumorigenesis rather than in the disease progression. Overall, this is an interesting paper to provide the evidence of the potential relevance of NANOG in OSCC formation, especially the examination of NANOG expression in 12 of 55 patients developed an invasive OSCC. However, there are several major issues that the authors need to address:

Response: We thank the reviewer for considering that this is an interesting paper, as it indeed provides first evidence for the relevant role of NANOG in early stages of OSCC tumorigenesis.     

Point 1: The scoring system of IHC used in this study is not convincing. The intensity of staining is not included and considered. The NANOG expression in tissues has to be evaluated with better scoring system, such as H-score system.

Response 1: A recent paper by Rodrigo et al. [16] uncovering an unprecedented role for NANOG as cancer risk marker prompted us to investigate its possible clinical significance in oral tumorigenesis. In an attempt to confirm these promising findings, we performed our study using the very same methodology, anti-NANOG antibody and IHC conditions originally described by Rodrigo et al. Accordingly, the same scoring system was applied in order to reproduce and obtain comparable results between both studies. In addition, we must clarify that staining intensity was indeed considered. As mentioned in Methods (lines 123-126) “A semiquantitative scoring system based on staining intensity was applied, as previously established [16], divided into three categories: negative (absence of staining, score 0), weak to moderate (some cytoplasmic staining in dysplastic areas, score 1), and strong protein expression (intense and homogeneous cytoplasmic staining in dysplastic areas, score 2)”. Furthermore, analogous scoring system based on staining intensity was applied in OSCC, as negative (0), weak (1) and strong protein expression (2), as previously established [Piazzolla et al. Nat Comm 2014, now Ref. 18]. This is also mentioned in Methods (lines 128-130).

Point 2: In Table 1, the total patient numbers of several characteristics are not 55. If the information is not available, it has to be noted below.

Response 2: We fully agree. The reviewer is correct. Regarding tobacco and alcohol consumption, information was only available for 31 patients, and ten of these patients (32%) were smokers and 4 (13%) habitual alcohol drinkers. This is now precisely indicated in Results section “3.1. Patient characteristics” (lines 145-146). Table 1 (footnotes) has also been accordingly amended.

Point 3: The authors conclude that the expression of NANOG in OPMD tissues is correlated with progression of OPMD to carcinoma (Table 3). However, for 12 OSCC tissues derived from the OPMDs, only 5 OPMD tissues express the NANOG proteins (Table 2). And the NANOG protein can be detected in only 31% of OSCC tissues (Table 5), suggesting that the expression of NANOG may play a minor role in progression of OPMD to carcinoma. This likely explains that only the dysplasia grading was significant independent predictor of oral cancer development in the multivariate Cox analysis (Table 4), while one would expect the NANOG level is also significant.

Response 3: In fact, NANOG expression in OPMD significantly correlated with increased risk of progression to invasive carcinoma, as consistently shown in Table 3 and also Kaplan-Meier cancer-free survival curves for both cytoplasmic and nuclear NANOG in Figure 2 (Figure 1 in the previous version of the manuscript). Nevertheless, we agree that since NANOG expression was only detected in 5 out 12 OPMDs that subsequently progressed to carcinoma, this suggests that NANOG seems to partially contribute as a driver gene to promote OSCC tumorigenesis. Alternatively, spatial-temporal reasons could also explain the lack of NANOG expression in OPMD that lately progressed to OSCC, as plausibly, OPMDs could have been biopsied before aberrant NANOG expression occurred, or cancer could develop from lesions not clinically visible at the time of biopsy and consequently unexamined. This has now been discussed (lines 272-277).

It is also worth noting that although histopathological grading was the only significant independent predictor of oral cancer development in the multivariate Cox analysis, patients harboring lesions with strong NANOG expression also clearly showed a higher risk of progression (HR >4), almost reaching statistical significance (P = 0.07), as already discussed (lines 269-272). Despite dysplasia grading still remains the gold standard in clinical practice for cancer risk assessment and decision-making and it was the most robust predictor of cancer risk in our series of OPMDs, accumulating evidences indicate that this grading system is largely subjective and affected by a great inter-examiner and intra-examiner variability. Noteworthy, we have used the new binary grading system (high-grade vs low-grade) proposed in the 2017 fourth edition of the WHO Classification (Ref. 17), as an alternative to overcome the still limited predictability of the previous WHO dysplasia grade into three categories: mild, moderate and severe dysplasia/CIS.

We have previously reported the identification of various cancer risk biomarkers that exhibited higher predictability beyond dysplasia grading, such as CTTN, FAK and Podoplanin that were significant independent predictors of oral cancer risk but not histopathological diagnosis (new Ref. 24, 25). Interestingly, Podoplanin (PDPN) has been identified as a marker for tumor-initiating cells (TIC) in squamous cell carcinomas (32). However, remarkably, the WHO three-tier grading was used in all these studies, and this could also play a role in the differences of predictive values observed with the present study. This information is now included in the Discussion (lines 282-289).

New References:

de Vicente JC, Rodrigo JP, Rodriguez-Santamarta T, Lequerica-Fernández P, Allonca E, García-Pedrero JM. Cortactin and focal adhesion kinase as predictors of cancer risk in patients with premalignant oral epithelial lesions. Oral Oncol. 2012, 48, 641-646, doi: 10.1016/j.oraloncology.2012.02.004. de Vicente JC, Rodrigo JP, Rodriguez-Santamarta T, Lequerica-Fernández P, Allonca E, García-Pedrero JM. Podoplanin expression in oral leukoplakia: tumorigenic role. Oral Oncol. 2013, 49, 598-603, doi: 10.1016/j.oraloncology.2013.02.008. Atsumi, N.; Ishii, G.; Kojima, M.; Sanada, M.; Fujii, S.; Ochiai, A. Podoplanin, a novel marker of tumor-initiating cells in human squamous cell carcinoma A431. Biochem Biophys Res Commun 2008, 373, 36-41, doi: 10.1016/j.bbrc.2008.05.163.

Point 4: The rationale for selecting NANOG in OSCC research may be not sufficient. If the NANOG is selected based on the cancer stem cell (CSC) hypothesis, the survey of other CSC-related molecules in OPMD and/or OSCC tissues should be included in the study.

Response 4: Following the reviewer’s suggestion, we have further extended our data by analyzing the expression of NANOG and other CSC-related genes (i.e. OCT4, SOX2 and PDPN) using RNAseq data in a subset of 172 OSCC patients from the TCGA cohort (Nature 2015, new Ref. 19) and the platforms cBioportal and UALCAN (new Refs. 20, 21). This information has been included in new Figure 3 and Results (lines 232-263). Moreover, the correlation between NANOG and PDPN protein expression was also assessed in 26 OPMDs with data available (Supplementary Information Table S2 and Results lines 251-254). We found a significant inverse association between NANOG and PDPN proteins in OPMDs (Chi square test, P = 0.017).

These new data provided valuable mechanistic information, as it follows: i) NANOG mRNA up-regulation is detected in OSCC although at much lower frequency than NANOG protein expression suggesting the involvement of post-transcriptional mechanisms rather than OCT4-dependent transcriptional regulation; ii) patients carrying NANOG mRNA up-regulation exhibited higher survival, as likewise observed for the patients with NANOG-positive expression in our OSCC cohort; iii) mRNA levels of other CSC-related genes OCT4, SOX2 and PDPN were found to be consistently up-regulated in OSCC patients, although these alterations did not overlap; iv) NANOG and PDPN protein expression in OPMDs also showed no overlap despite the expression of each protein significantly predicted increased risk of malignant progression, thereby suggesting an independent role of these proteins as CSC or TIC during oral carcinogenesis. This is now discussed (lines 383-394).

Recent advances in NGS and omics technologies have enormously contributed to uncover the high complexity and heterogeneity of oncogenomes (new Refs. 26, 27). Beyond the great diversity of genetic and epigenetic alterations found within a tumor, the interaction with the surrounding microenvironment also dynamically modulates the tumor heterogeneity (new Ref. 28). In addition, epithelia to mesenchymal transition and CSC plasticity has also been demonstrated to fuel tumor heterogeneity in response to environmental cues to drive tumor spreading and therapeutic resistance (new Refs. 29, 30). The fact that OSCC show a heterogeneous architecture compared with healthy oral mucosa has also led to the hypothesis that only a small clonogenic subpopulation of cells considered CSC or tumor-initiating cells (TICs) is responsible for generating tumors (new Ref. 31). In this regard, Podoplanin (PDPN) has been identified as a marker for tumor-initiating cells (TIC) in squamous cell carcinomas (new Ref. 32), and PDPN-positive cells beyond the basal layer of the oral epithelium have been interpreted as an upward clonal expansion of stem cells during carcinogenesis (new Ref. 33). Consistent with this role, PDPN-positive OPMDs harboring such clonal expansion exhibited a significantly higher risk of malignant transformation, as we and others demonstrated (new Refs. 25, 33, 34). This information has been added to the Discussion (lines 290-303).

The impact of cytoplasmic NANOG expression on the disease-specific survival was also assessed in the present study. Kaplan-Meier analysis showed a tendency, but not significant, between positive NANOG expression and better survival in our cohort of OSCC patients. We obtained analogous findings by analyzing NANOG mRNA levels in an independent cohort of 172 OSCC from the TCGA cohort. Thus, higher survival was also observed in patients harboring NANOG mRNA up-regulation. Consistent with this, we also found that positive NANOG expression was more frequent in pN0 tumors, early I-II stages, and absence of tumor recurrences. In line with these findings, high expression of OCT4 and SOX2 has been associated with earlier stage, small tumor size, and the absence of lymph node metastasis, and high SOX2 expression was significantly associated with better disease-specific survival in OSCC [1]. This has now been included in the Discussion (lines 361-367).

New References:

Cancer Genome Atlas Network. Comprehensive genomic characterization of head and neck squamous cell carcinomas. Nature 2015, 517, 576–582, doi: 10.1038/nature14129. Cerami, E.; Gao, J.; Dogrusoz, U.; Gross, B.E.; Sumer, S.O.; Aksoy, B.A.; Jacobsen, A.; Byrne, C.J.; Heuer, M.L.; Larsson, E.; et al. The cBio cancer genomics portal: An open platform for exploring multidimensional cancer genomics data. Cancer Discov. 2012, 2, 401–404, doi: 10.1158/2159-8290.CD-12-0095. Chandrashekar, D.S.; Bashel, B.; Balasubramanya, S.A.H.; Creighton, C.J.; Rodriguez, I.P.; Chakravarthi, B.V.S.K.; Varambally, S. UALCAN: A portal for facilitating tumor subgroup gene expression and survival analyses. Neoplasia. 2017, 19, 649-658, doi: 10.1016/j.neo.2017.05.002. de Vicente JC, Rodrigo JP, Rodriguez-Santamarta T, Lequerica-Fernández P, Allonca E, García-Pedrero JM. Podoplanin expression in oral leukoplakia: tumorigenic role. Oral Oncol. 2013, 49, 598-603, doi: 10.1016/j.oraloncology.2013.02.008. Stransky, N.; Egloff, A.M.; Tward, A.D.; Kostic, A.D.; Cibulskis, K.; Sivachenko, A.; Kryukov, G.V.; Lawrence, M.S.; Sougnez, C.; McKenna, A.; et al. The mutational landscape of head and neck squamous cell carcinoma. Science 2011, 333, 1157-1160, doi: 10.1126/science.1208130. Hedberg, M.L.; Goh, G.; Chiosea, S.I.; Bauman, J.E.; Freilino, M.L.; Zeng, Y.; Wang, L.; Diergaarde, B.B.; Gooding, W.E.; Lui, V.W.; et al. Genetic landscape of metastatic and recurrent head and neck squamous cell carcinoma. J Clin Invest 2016, 126, 1606, doi: 10.1172/JCI86862. Álvarez-Teijeiro, S.; García-Inclán, C.; Villaronga, M.Á.; Casado, P.; Hermida-Prado, F.; Granda-Díaz, R.; Rodrigo, J.P.; Calvo, F.; Del-Río-Ibisate, N.; Gandarillas, A.; et al. Factors Secreted by Cancer-Associated Fibroblasts that Sustain Cancer Stem Properties in Head and Neck Squamous Carcinoma Cells as Potential Therapeutic Targets. Cancers (Basel). 2018, 10, pii: E334, doi: 10.3390/cancers10090334. Prasetyanti, P.R.; Medema, J.P. Intra-tumor heterogeneity from a cancer stem cell perspective. Mol Cancer 2017, 16, 41, doi: 10.1186/s12943-017-0600-4. Biddle, A.; Gammon, L.; Liang, X.; Costea, D.E.; Mackenzie, I.C. Phenotypic Plasticity Determines Cancer Stem Cell Therapeutic Resistance in Oral Squamous Cell Carcinoma. EBioMedicine 2016, 4, 138-145, doi: 10.1016/j.ebiom.2016.01.007. Al-Hajj, M.; Clarke, M.F. Self-renewal and solid tumor stem cells. Oncogene 2004, 23, 7274-7282, doi: 10.1038/sj.onc.1207947. Atsumi, N.; Ishii, G.; Kojima, M.; Sanada, M.; Fujii, S.; Ochiai, A. Podoplanin, a novel marker of tumor-initiating cells in human squamous cell carcinoma A431. Biochem Biophys Res Commun 2008, 373, 36-41, doi: 10.1016/j.bbrc.2008.05.163. Kawaguchi, H.; El-Naggar, A.K.; Papadimitrakopoulou, V.; Ren, H.; Fan, Y.H.; Feng, L.; Lee, J.J.; Kim, E.; Hong, W.K.; Lippman, S.M.; et al. Podoplanin: A novel marker for oral cancer risk in patients with oral premalignancy. J Clin Oncol 2008, 26, 354-360, doi: 10.1200/JCO.2007.13.4072. Rodrigo, J.P.; García-Carracedo, D.; González, M.V.; Mancebo, G.; Fresno, M.F.; García-Pedrero, J. Podoplanin expression in the development and progression of laryngeal squamous cell carcinomas. Mol Cancer 2010, 9, 48, doi: 10.1186/1476-4598-9-48.

Round 2

Reviewer 2 Report

The manuscript has been considerably improved. However, the main drawback of the work still remains. In this manuscript, all of the results were exclusively based on the MANOG IHC. There is no any additional experiments that demonstrate the role of MANOG in generation and/or progression of oral cancers, indicating that the analysis of IHC results is the most important for the manuscript. The scoring system of IHC used in this study is not convincing enough, although the authors have listed the references that stand for the application of the method used here. The intensity of staining is not included and considered. The NANOG expression in tissues has to be evaluated with better scoring system, such as H-score system.

Author Response

Reviewer #2

Comments and Suggestions for Authors

The manuscript has been considerably improved.

Response: We thank the reviewer for considering that the additional data provided in our revised version considerably and satisfactorily improved the manuscript. All his/her valuable comments and insightful recommendations were indeed highly appreciated.

Point 1: However, the main drawback of the work still remains. In this manuscript, all of the results were exclusively based on the MANOG IHC. There is no any additional experiments that demonstrate the role of MANOG in generation and/or progression of oral cancers, indicating that the analysis of IHC results is the most important for the manuscript.

Response 1: The present study was conducted to comprehensively investigate the clinical relevance of NANOG expression in both early stages of oral carcinogenesis and late stages of disease progression to establish correlations with the risk of progression to oral cancer, impact on OSCC prognosis and patient outcome, as precisely indicated in the last paragraph of the Introduction. NANOG protein expression was thus analyzed in large series of oral dysplastic lesions together with OSCC tissue specimens from the same institution, using the very same methodology, IHC conditions and scoring to obtain comparable data. Quite remarkably, this study is the first to demonstrate the early occurrence and clinically relevant role of NANOG expression in oral tumorigenesis rather than in late stages of OSCC progression, and also first to uncover the potential application of NANOG expression as an early predictor of oral cancer risk in patients with OPMDs. In fact, this was previously highlighted by the reviewer “Overall, this is an interesting paper to provide the evidence of the potential relevance of NANOG in OSCC formation, especially the examination of NANOG expression in 12 of 55 patients developed an invasive OSCC”.

In addition, our in silico analysis of the transcriptome data from the TCGA further and significantly contributed to provide valuable mechanistic information regarding the up-regulation of NANOG mRNA expression and other CSC-related genes in OSCC patients. The major findings have been detailed in the last paragraph of Discussion.

We truly think this is a clinical translationally-oriented research work that adequately fits into the special Issue "Advances in Clinical and Translational Research of Oral Surgery, Biomaterials, and Oral Disease Management". In this sense, since immunohistochemical NANOG evaluation is a relatively simple and objective method that could be easily implemented in the clinical practice and widely affordable in almost any hospital. A vast amount of diagnostic tests have been developed based on the detection of IHC markers, which have emerged as valuable tools in diagnostic pathology for multiple diseases, and still are highly and widely used in clinical routine.

On the other hand, the pathobiological role of NANOG has already been investigated in HNSCC and other cancers, thereby demonstrating its contribution to tumor formation and progression and potential as a therapeutic target in various in vitro and in vivo models, as in the references below.

Palla AR, Piazzolla D, Alcazar N, Cañamero M, Graña O, Gómez-López G, Dominguez O, Dueñas M, Paramio JM, Serrano M. The pluripotency factor NANOG promotes the formation of squamous cell carcinomas. Sci Rep. 2015 May 19;5:10205. doi: 10.1038/srep10205.

Piazzolla D, Palla AR, Pantoja C, Cañamero M, de Castro IP, Ortega S, Gómez-López G, Dominguez O, Megías D, Roncador G, Luque-Garcia JL, Fernandez-Tresguerres B, Fernandez AF, Fraga MF, Rodriguez-Justo M, Manzanares M, Sánchez-Carbayo M, García-Pedrero JM, Rodrigo JP, Malumbres M, Serrano M. Lineage-restricted function of the pluripotency factor NANOG in stratified epithelia. Nat Commun. 2014 Jun 30;5:4226. doi: 10.1038/ncomms5226.

Xie X, Piao L, Cavey GS, Old M, Teknos TN, Mapp AK, Pan Q. Phosphorylation of Nanog is essential to regulate Bmi1 and promote tumorigenesis. Oncogene. 2014 Apr 17;33(16):2040-52. doi: 10.1038/onc.2013.173.

Huang CE, Yu CC, Hu FW, Chou MY, Tsai LL. Enhanced chemosensitivity by targeting Nanog in head and neck squamous cell carcinomas. Int J Mol Sci. 2014 Aug 25;15(9):14935-48. doi: 10.3390/ijms150914935.

Lee SH, Nam HJ, Kang HJ, Kwon HW, Lim YC. Epigallocatechin-3-gallate attenuates head and neck cancer stem cell traits through suppression of Notch pathway. Eur J Cancer. 2013 Oct;49(15):3210-8. doi: 10.1016/j.ejca.2013.06.025.

Yu MA, Kiang A, Wang-Rodriguez J, Rahimy E, Haas M, Yu V, Ellies LG, Chen J, Fan JB, Brumund KT, Weisman RA, Ongkeko WM. Nicotine promotes acquisition of stem cell and epithelial-to-mesenchymal properties in head and neck squamous cell carcinoma. PLoS One. 2012;7(12):e51967. doi: 10.1371/journal.pone.0051967.

This reference has been added to the Discussion as new Ref 48, as it supports our results indicating a link between NANOG expression and tobacco consumption in OSCC patients.

Point 2: The scoring system of IHC used in this study is not convincing enough, although the authors have listed the references that stand for the application of the method used here. The intensity of staining is not included and considered. The NANOG expression in tissues has to be evaluated with better scoring system, such as H-score system.

Response 2: We justified the scoring system used as that previously established in two highly relevant and closely-related papers (Refs 16 and 18). Please note that these data are not incompatible with the application of a H-score system, which normally involves jointly the evaluation of both staining intensity and percentage of stained cells. Given that CSC subpopulations could represent a very small percentage of cells, hence NANOG expression in few cells even as low as 1 % could be intrinsically meaningful into the CSC concept. Taking this into consideration, any NANOG-positive cell was considered even 1% of positive cells. As a result, a semiquantitative scoring system based on staining intensity was applied, divided into three categories: negative (absence of staining, score 0), weak to moderate (score 1), and strong protein expression (score 2). Again, since any NANOG-positive cell could be meaningful, this criteria was used as a cut-off point to establish positive NANOG expression (scores > 0) versus negative expression (score = 0). If a H-score system were used e.g. considering staining intensity (graded from 0 to 2) and the % of positive cells in a range from 0 to 200, the same criteria would be applied and data dichotomized as positive NANOG expression (scores > 0) versus negative expression (score = 0). Consequently, the scoring system applied in this particular case for this protein will not vary our results. Further details have been included in Methods (lines 123-126 and 133-135) to clarify this.